# TIM: INTERPRETABLE MODELLING OF COMPLEX TEMPORAL INTERACTIONS IN MULTIVARIATE NETWORKS

## ABSTRACT

Multivariate time series forecasting is crucial across various fields and essential for addressing numerous real-world challenges. However, existing forecasting methods have significant limitations: while Transformer models are effective, they are constrained by high computational costs and declining performance in long-term forecasting; MLP models struggle to capture complex multivariate interactions. These issues hinder the models' ability to accurately decompose seasonality and trends. To tackle these problems, we propose a new method called TIM. Through a cross-layer architecture, TIM decomposes time series predictions into temporal features, multivariate interaction features, and residual components. Our all-MLP model integrates global features with complex multivariate dynamics. By introducing a linear self-attention mechanism across variables and time steps, TIM enhances the learning of feature interactions and accurately captures temporal transitions between domains. This innovative design leverages linear attention mechanisms and cross-layer architecture to more effectively model temporal features and multivariate interactions. It surpasses traditional Transformer-based methods by improving predictive accuracy while maintaining linear computational complexity. Experimental results demonstrate that TIM outperforms existing state-of-the-art methods while ensuring computational efficiency.

## 1 INTODUCTION

Long sequence time-series forecasting (LSTF) is essential across various industries, including weather forecasting (Ahamed & Cheng (2024)), traffic volume prediction (Zhao (2019)), electricity transformer temperature monitoring (Zhou et al. (2021)), and electric power consumption (Hebrail & Berard (2006)). Transformers, with their innovative attention mechanisms, have made significant strides in time series forecasting by capturing complex dependencies and multi-level representations from sequential data. Despite these advancements, Transformers are often hampered by high computational costs and performance degradation over longer sequences.

Recent developments in deep learning have introduced several models designed to enhance time series forecasting, including Transformers (Lim et al. (2021); Liu et al. (2024); Zhang et al. (2024a)), RNNs (Damaševičius et al. (2024); De et al. (2024)), SSMs (Rangapuram et al. (2018); Auger-Méthé et al. (2021); Newman et al. (2023); Orvieto et al. (2023)), and MLPs (Yi et al. (2024); Zhang et al. (2022); Yeh et al. (2024); Zeng et al. (2023)). While Transformer-based solutions have achieved notable results, they often do not significantly outperform linear models when accounting for the computational overhead associated with their increased parameter volumes. The quadratic complexity of

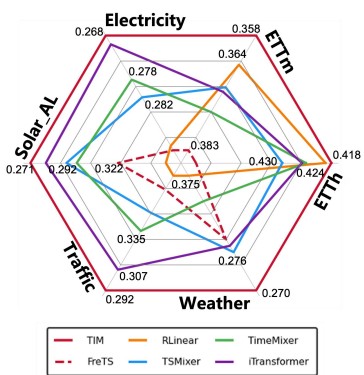

Figure 1: Average MAE performance of TIM. Model performance is derived from our reimplemented experimental results.

Transformers, scaling with the context length, poses significant scalability challenges, especially for long sequences. Research indicates that linear models can sometimes be more effective and efficient for time series forecasting (Zeng et al. (2023)).

In developing advanced forecasting architectures, several approaches have been employed, including series decomposition (Wu et al. (2021); Zhou et al. (2022); Bandara et al. (2020); Hao & Liu (2024)) and channel-independent (CI) versus channel-dependent (CD) methods (Liang et al. (2023); Nie et al. (2023; 2024)). However, these methods often face limitations due to non-stationarity, evolving seasonal variations, and uncertainties in trend identification. Data acquisition issues, such as sensor inaccuracies, further complicate effective time series modelling.

Addressing these issues, we introduce **TIM**, a groundbreaking approach that enhances long sequence time-series forecasting by leveraging a purely Multi-Layer Perceptron (MLP)-based architecture. Our model innovatively integrates linear attention and cross-layer mechanisms to tackle the inherent limitations of existing methods. Specifically, **TIM** features:

- **Efficiency and Scalability: TIM** achieves competitive forecasting performance with linear complexity and fewer parameters, significantly improving efficiency compared to traditional Transformer-based models, which suffer from quadratic complexity.
- **Enhanced Multivariate Interaction Modeling:** Unlike traditional MLPs, **TIM** excels at capturing complex multivariate interactions. Our cross-layer design effectively models intricate dependencies between multiple variables, addressing the limitations of existing MLP approaches in handling multivariate data.
- **Interpretability and Robustness: TIM** incorporates mechanisms that enhance interpretability while providing robust performance across real-world time series data. By integrating independent feature processing with correlated channel interactions, **TIM** not only improves prediction accuracy but also offers insights into how different features and interactions contribute to the forecasting results.

Our approach demonstrates superior forecasting accuracy and computational efficiency compared to current state-of-the-art methods, offering a robust and scalable solution for complex time series forecasting tasks.

## 2    RELATED WORK

### 2.1    PROBLEM STATEMENT

In the context of multivariate time series analysis, let $X = \{x_1^{(c)}, \ldots, x_L^{(c)}\}_{f=1}^F$ denote a collection of $F$ feature channels, where each channel $c$ comprises an independent sequence of $L$ observations within a look-back window. The channel index $f$ will be omitted in subsequent discussions for simplicity. The objective of the forecasting task is to predict the future values of the time series over the next $pred\_len$ time steps, denoted as $\hat{X}_{L+1:L+P}$, based on the historical data $X_{1:L}$, where pred_len is abbreviated as $P$. This prediction is achieved through a forecasting function $F(\cdot)$, which is instantiated as an MLP-based model in this study. Our primary goal is to mitigate the high computational cost and performance degradation associated with long-term data and to enhance model prediction capabilities through multivariable feature interaction and long-term series distribution migration modelling. This approach seeks to improve the forecasting outcome $X'$, specifically by minimizing the error between the predicted values $X'$ (i.e., $F(X_{1:L})$) and the true future values $\hat{X}_{L+1:L+P}$. Traditionally, time series data are usually subjected to batch normalization before being input into prediction models. However, recent research has highlighted the efficacy of utilizing a reversible instance normalization (RevIN: Kim et al. (2022)) in addressing the challenges posed by distribution shifts in time-series forecasting problems.

### 2.2    TEMPORAL MODELING FOR LSTF

In the realm of Long Short-Term Forecasting (LSTF) tasks, Transformer-based and MLP-based models have emerged as the preeminent backbones due to their exceptional temporal modelling capabilities. Deviating from the Vanilla Transformer (Ashish (2017)), recent research has advanced

the field significantly. Notably, Informer (Zhou et al. (2021)) introduced an innovative strategy whereby timestamps are encoded as supplementary positional encodings through the deployment of learnable embedding layers. This advancement, along with subsequent works such as Autoformer (Wu et al. (2021)) and FEDformer (Zhou et al. (2022)), has firmly established these foundational architectures as widely acknowledged solutions for addressing LSTF challenges. Subsequent endeavours have introduced iTransformer, a variant that ingeniously applies the attention mechanism and feed-forward network on inverted dimensions. This innovation not only diversifies the Transformer family but also propels its performance to new heights, further demonstrating the potential and adaptability of Transformer-based models in handling complex tasks. Furthermore, the MLPs (Oreshkin et al. (2019); Challu et al. (2023)) achieve favourable performance in both forecasting performance and efficiency for LSTF tasks. Previous research has demonstrated that MLPs can achieve the same top level of performance as Transformers in long-term sequential forecasting tasks using trend season decomposition methods (Zeng et al. (2023)). Recent research on TimeMixer (Wang et al. (2024)) has elegantly capitalized on disentangled multiscale series, leveraging them effectively in both the past extraction and future prediction phases. This approach has demonstrated remarkable achievements, consistently attaining state-of-the-art performances across both long-term and short-term forecasting tasks, while also exhibiting favourable run-time efficiency, underscoring its practical significance and efficiency in real-world applications.

Traditional sequential models, such as Recurrent Neural Networks (RNNs), frequently encounter issues of gradient vanishing or gradient explosion when dealing with long time series, rendering them challenged in capturing long-range dependencies. The Attention mechanism, by directly computing the relevance between any two positions within the sequence, can mitigate this problem to some extent, enabling the model to process long sequence data more effectively. By incorporating the Attention mechanism, the model is able to dynamically allocate more importance or "focus" on the most relevant parts of the input sequence, regardless of their positions within the sequence. The following equation can formulate the classic attention mechanism, particularly within the framework of self-attention or transformer-based models, Q typically represents the "Query", K denotes the "Key", and V stands for the "Value". we ignore the normalization term for simplicity.

$$\text{Attention}(Q, K, V) = \text{softmax}(QK^T)V \tag{1}$$

In classical attention mechanisms, both spatial and temporal complexities scale with $O(n^2)$, where n represents the sequence length. Consequently, as n increases significantly, the computational burden on Transformer models becomes prohibitively high. Recently, extensive research has focused on addressing this issue by reducing the computational cost of Transformer models. These efforts include various techniques such as Sparse Attention (Wu et al. (2020); Zhang et al. (2024b)), and quantization. Additionally, modifications to the attention architecture have been explored to reduce its complexity to $O(n \log(n))$ or even $O(n)$, thereby improving the scalability and efficiency of Transformer models for processing longer sequences.

## 2.3 LINEAR ATTENTION

The Attention mechanism of equation 1 can be rewritten in the following way:

$$\text{Attention}(Q, K, V)_i = \frac{\sum_{j=1}^{n} \exp\left(q_i^\top k_j\right) v_j}{\sum_{j=1}^{n} \exp\left(q_i^\top k_j\right)} = \frac{\sum_{j=1}^{n} \text{sim}(q_i, k_j) v_j}{\sum_{j=1}^{n} \text{sim}(q_i, k_j)} \tag{2}$$

Previous research (Wang et al. (2018)) had pointed out that if we use $\text{sim}(q_i, k_j) = \phi(q_i)^\top \varphi(k_j)$ to simplify the calculation of attention, then the complexity problem of attention mechanism should be mitigated. $\phi(x), \varphi(x)$ are defined as $\phi(x) = \varphi(x) = elu(x) + 1$, where $elu(x)$ denotes the Exponential Linear Unit (as introduced by Clevert (2015)). The additional "+1" term ensures that the similarity term remains positive. From the perspective of the result, equation 2 expresses that the core logic of the attention mechanism lies in focusing on everything and the key points. It can be seen from the weighted sum expression of the Attention formula that the self-attention mechanism can help to model the entire time series and automatically help the model focus on the local feature.

In our work, we harness the merits of the linear attention mechanism to explicitly model the multi-variable interaction across the entire time series of individual variables, as well as the evolving features within cross-sectional multi-variable data. This approach endows our model with several

advantageous characteristics, including reduced computational complexity, minimized storage requirements, the capability to model the global time series, localized feature attention, and the proficiency to handle multi-variable relationships. We will delve deeper into the intricate architecture of our model in the subsequent method section.

### 2.4 FEATURE FUSION

To leverage linear attention effectively in capturing both the multi-variable interactions across the entire time series of individual variables and the evolving features within cross-sectional multi-variable data, our approach aims to extract meaningful global information from the time series and accurately represent the intricate multi-variable relationships. This process is non-trivial and frequently necessitates intricate manual feature engineering or an exhaustive search procedure. Previous work Wang et al. (2017) introduces a novel cross-network that is more efficient in learning certain bounded-degree feature interactions when it keeps the benefits of MLPs without extra complexity. This enables our model to comprehensively analyze and understand the dynamics within and across variables over time.

## 3 TIM

### 3.1 GENERAL ARCHITECTURE

According to Li et al. (2023), our model, like many others, consists of three key components: RevIN, a reversible normalization layer; an MLP; and a linear projection layer that generates the final prediction results. In our proposed architecture, MLP is used to extract time series features. In subsequent modules, we will employ a decomposition method to enable our model to learn from multivariate interaction features, temporal characteristics of the time series, and decomposed components, respectively. The full architecture of TIM can be found in Figure 2.

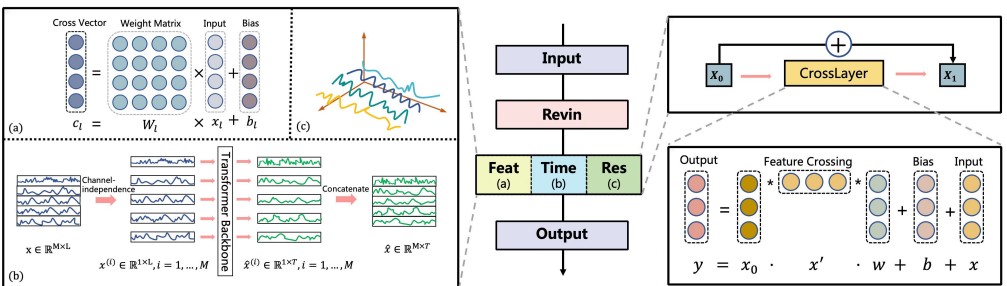

Figure 2: Overall TIM Architecture. TIM consists of three key components: Feat Fusion, which extracts multivariate interaction features; Time Fusion, which models temporal shifts across time points; and a residual modelling component for temporal, multivariable, or noise effects. The outputs of these modules—$X_{Feat}$, $X_{Time}$, and $X_{Res}$—are combined to produce the final forecast, which is then passed through a linear projection layer and inverse-transformed via RevIN to scale it back to the target domain for the prediction horizon.

### 3.2 FUSION ARCHITECTURE IN TIM

In the current state-of-the-art approaches for Long Sequence Time Forecasting (LSTF), many works have leveraged decomposition methods to enhance model performance. However, no existing research has yet explored decomposing long time series into univariate time series and single time snapshots. In the Deep Cross Network (DCN) paper, the authors employed a highly efficient and indirect method to achieve explicit feature crossing. This technique lays the foundation for our innovative approach to decompose long time series into univariate time series and single time snapshots, while simultaneously capturing both multivariate interaction features and temporal characteristics.

In previous research efforts, a significant body of work Bandara et al. (2020); Hao & Liu (2024); Wu et al. (2021); Zeng et al. (2023) has utilized seasonal and trend decomposition techniques to

enhance model performance in long-term time series analysis. These methods decompose data into distinct seasonal components $s(t)$ and trend components $f(t)$, while managing acceptable levels of noise, thus improving overall predictive capabilities. Although these decomposition techniques have proven effective for both MLP-based and Transformer-based models in Long Sequence Time Forecasting (LSTF) tasks, we contend their general applicability is limited.

In the context of long temporal sequences, the complexity of the data can lead to extreme imbalances between trend or seasonal components and the residual (noise) component. When the magnitude of one component becomes comparable to that of the residual, traditional decomposition methods, such as moving averages, may inadvertently capture noise as part of the trend or seasonal components. This issue is particularly pronounced when dealing with rapidly changing components, as these methods struggle to adapt to such volatile elements.

To address these challenges, we propose a novel approach that decomposes the model into three main components. The first component, processed through the Feat_Fusion module, extracts multivariate interaction features from the time series. The second component models the explicit temporal shifts of multivariate features at individual time points using single time snapshots, which are then processed by the Time_Fusion module to capture temporal shift characteristics across time nodes. The primary difference between the Time_Fusion and Feat_Fusion modules lies in their input and output dimensions due to matrix transfer, although they share the same underlying structure. The features $X$ are compared with those obtained from $X\_Feat$ and $X\_Time$, and the residuals are treated as potential seasonal, trend, or noise components. These residuals are modelled via a network structure analogous to the Time_Fusion module, resulting in $X_{Res}$. The final output is computed as $Y = X_{Feat} + X_{Time} + X_{Res}$, which is then passed through a linear projection layer to produce the time series forecast for the prediction horizon. Finally, the output is inverse-transformed via the RevIN layer to scale it back to the target domain.

### 3.3 LINEAR ATTENTION GATED UNIT FOR FEATURE EXTRACTION

In this section, we will provide a detailed analysis of the Time_Fusion, Feat_Fusion, and Res_Fusion modules used for extracting time series features. The primary distinction among these three modules lies in their input-output architecture, while they all share the same feature extraction algorithm. Both Time_Fusion and Res_Fusion have identical input and output dimensions, with their input-output dimensions given by$\in \mathbb{R}^{F \times H}$. The input-output dimensions of Feat_Fusion $\in \mathbb{R}^{F \times H}$

---

**Algorithm 1** Fusion Architecture for Time_Fusion, Feat_Fusion and Res_Fusion

---

**Require:** Input $X_0 \in \mathbb{R}^{F \times H}$. Number of Layers $N$. Sigmoid function denoted as $\sigma$. Concatenate function denoted as cat. Linear layer mappings from the dimension 2*dim to dim, denoted as combine and gate.

**Ensure:** Output $X_L \in \mathbb{R}^{F \times H}$

1: Initialize $X_i = X_0$
2: **for** $i = 1$ to $N$ **do**
3:     Compute $y = W_i X_i + b_1$ $\{y \in \mathbb{R}^{F \times H}\}$
4:     Compute residual res $= \tilde{W}_i(X_i - y) + b_2$ $\{\text{res} \in \mathbb{R}^{F \times H}\}$
5:     Concatenate $x_{\text{cat}} = \text{cat}(y, \text{res})$ $\{x_{\text{cat}} \in \mathbb{R}^{F \times 2H}\}$
6:     Apply ELU activation $x_{\text{elu}} = \text{ELU}(x_{\text{cat}}) + 1$
7:     Gate and combine $x_{\text{out}} = \text{combine}(x_{\text{elu}}) \cdot \sigma(\text{gate}(x_{\text{elu}}))$
8:     Update $X_1 = X_0 \cdot x_{\text{out}}$ $\{x_1 \in \mathbb{R}^{F \times H}\}$
9:     Apply Dropout $X_1 = \text{Dropout}(X_1)$
10:     Update $X_i = X_i + X_1$ $\{x_i \in \mathbb{R}^{F \times H}\}$
11: **end for**
12: **return** $X_i$

---

Time_Fusion, Feat_Fusion and Res_Fusion share the same architecture described in Algorithm 1. The design objective of the Time-Fusion module lies in leveraging the concept of linear attention to facilitate the model's capability to learn features from temporal sequences, as evidenced through a series of mathematical derivations. In the context of linear attention mechanisms, weights are typically derived by computing the similarity between Query and Key vectors. However, in this particular implementation, the weights are obtained through an element-wise multiplication operation

with the initial input $X_0$. The proposed approach enables our model to achieve linear self-attention and progressively transfers the temporal sequences from the latent space of the source domain into the state space of the target domain. In the absence of residual connections, the algorithm can be succinctly expressed by the following equation, where $\circ$ is the Hadamard Product (point-wise multiplication) and $D$ stands for the dropout layer:

$$X_N = X_0 + \sum_{i=1}^{N} D(X_0 \circ (ELU(W_i X_{i-1} + b_i) + 1)) \tag{3}$$

Dropout can be seen as an implicit gating mechanism that randomly discards a part of neurons, similar to the suppression of irrelevant information in the attention mechanism. Although it does not explicitly use gating operations, it is similar in effect to the attention weight distribution in the attention mechanism. To make the model more sensitive to state changes, we added and designed a residual structure to help the model better capture temporal state transitions. In Algorithm 1, the dimensions leveraged within the Time_fusion and Res_fusion modules are preserved consistently. However, within the Feat_fusion module, a crucial transformation occurs before the module's input, where matrices undergo a transposition. Consequently, within the Feat_fusion module, the residual structure operates along the feature dimension, F, effectively expanding the dimensionality from $H \times F$ to $H \times 2F$. Despite this reconfiguration, the self-attention mechanism within the module remains efficacious, now engaging in the learning process across multivariate features at each temporal node, facilitating an intricate understanding of the interdependencies within the feature space.

### 3.4 OVERALL-ARCHITECTURE OF TIM

Having delved into the intricacies of each module within our novel feature/time/resolution decomposition paradigm in the preceding section, we now present a concise summary of our model's overall workflow encapsulated in Algorithm 2. This summary provides a holistic view of how the individual components collaborate to perform their designated functions, offering a comprehensive understanding of our novel operational framework.

---

**Algorithm 2** TIM Overall Architecture

---

**Require:** Input lookback time series $X_{input} \in \mathbb{R}^{L \times F}$; input Length L ; predicted length P; variates number F ; hidden dimension H;
**Ensure:** $Y \in \mathbb{R}^{P*F}$
   $X \leftarrow Normalization(X)$
   $X \leftarrow Transpose(X_{input}) \; \{X \in \mathbb{R}^{F \times L}\}$
   $X \leftarrow Time\_Encoder(X) \; \{X \in \mathbb{R}^{F \times H}\}$
   $X_{time} \leftarrow Time\_Fusion(X)$
   $X_{feat} \leftarrow Transpose(Feat\_Fusion(Transpose(X)))$
   $X_{res} = Res\_Fusion(X - X_{feat} - X_{time})$
   $Y = X_{res} + X_{feat} + X_{time} \; \{Y \in \mathbb{R}^{F \times H}\}$
   $OUTPUT \leftarrow Proj(Y) \; \{OUTPUT \in \mathbb{R}^{F \times P}\}$
   $OUTPUT \leftarrow Transpose(OUTPUT)$
   $Prediction \leftarrow De - Normalization(OUTPUT)$
   **return** $Prediction \in \mathbb{R}^{P \times F}$

---

## 4 EXPERIMENTS

### 4.1 DATASET DESCRIPTION

We have conducted experiments on eight rigorously established benchmarks: the ETT datasets, which encompass four distinct subsets—ETTh1, ETTh2, ETTm1, and ETTm2—alongside Weather, Solar-Energy, Electricity, and Traffic datasets following Zhou et al. (2021); Zeng et al. (2023); Hebrail & Berard (2006); Zhao et al. (2019). These benchmarks serve as robust platforms for evaluating the performance and efficacy of our forecasting models in the long-term horizon.

## 4.2 Main Result

In our experimental setup for model evaluation, we have standardized the parameters across all models to ensure a fair comparison on a uniform platform. Specifically, we have fixed the input dimension to 96 and varied the prediction horizon for time series forecasting, encompassing lengths of [96, 192, 336, 720]. This approach allows for a comprehensive assessment of model performance under different forecasting scenarios. To measure various variables on a consistent scale, we compute the Mean Squared Error (MSE) and Mean Absolute Error (MAE) on the normalized data provided by Revin (Kim et al. (2021)). Additional details regarding the experimental settings, encompassing training specifics and hyperparameters, are furnished in the Appendix. The experiments were implemented using PyTorch (Paszke et al. (2019)) and executed on a single NVIDIA 4090 GPU with 24GB of memory.

For the smaller-scale datasets, such as ETT and Exchange, we have adopted a consistent set of hyperparameters to facilitate a rigorous comparison. Specifically, we have set the number of hidden layers (d_model) to 4, the number of encoder layers (e_layers) to 2, the dropout rate to 0.25, and the learning rate to 1e-3. These configurations have been chosen to balance model complexity and computational efficiency, aiming to achieve optimal performance on the specified datasets.

By adhering to these standardized parameters and experimental protocols, we aim to provide a robust and unbiased evaluation of the different models under investigation, enabling a more meaningful comparison of their strengths and limitations within the context of time series forecasting.

We select 7 SOTA baseline studies. We are focusing on both MLP-based and Transformer-based methods. We added DLinear (Zeng et al. (2023)), RLinear (Li et al. (2023)), TSMixer (Ekambaram et al. (2023)) and TimeMixer (Wang et al. (2024)). We also added PatchTST (Nie et al. (2023)) and iTransformer (Liu et al. (2024)).

Results of the main experiments can be found in Table 1,3. The optimal outcomes are emphasized in bold red font, while the second-best results are underscored in blue, facilitating a precise comparison of the performance levels achieved. Experimental studies have demonstrated that our model surpasses existing state-of-the-art (SOTA) methods, achieving SOTA performance in complex long-term time series forecasting tasks and multivariate prediction using a simple MLP model. We attribute these remarkable experimental results to our innovatively proposed time series decomposition framework, which concurrently addresses time series dynamics and multivariate interaction modelling. The hierarchical incorporation of a linear self-attention mechanism assists the model in capturing both temporal characteristics and multivariate interaction features, contributing to its outstanding performance.

Table 1: Multivariate forecasting results with prediction lengths in {96, 192, 336, 720} for eight benchmark datasets and fixed lookback length 96. Results are averaged from all prediction lengths. Avg means further averaged by subsets. Full results are listed in Table 3

| Models (Mean) | TIM Ours | | DLinear 2023 | | PatchTST 2023 | | FreTS 2024 | | RLinear 2023 | | TSMixer 2023 | | TimeMixer 2024 | | iTransFormer 2024 | | TimesNet 2023 | |
|---|---|---|---|---|---|---|---|---|---|---|---|---|---|---|---|---|---|---|---|
| Metric | mse | mae | mse | mae | mse | mae | mse | mae | mse | mae | mse | mae | mse | mae | mse | mae | mse | mae |
| ETTh1 | **0.434** | 0.433 | 0.452 | 0.447 | 0.440 | 0.442 | 0.464 | 0.447 | 0.443 | **0.431** | 0.456 | 0.446 | 0.465 | 0.450 | 0.448 | 0.443 | 0.531 | 0.491 |
| ETTh2 | 0.377 | 0.402 | 0.526 | 0.498 | 0.379 | 0.405 | 0.448 | 0.457 | 0.385 | 0.407 | 0.396 | 0.414 | **0.368** | **0.398** | 0.382 | 0.407 | 0.429 | 0.434 |
| ETTm1 | **0.382** | **0.397** | 0.404 | 0.408 | 0.444 | 0.457 | 0.432 | 0.438 | 0.409 | 0.400 | 0.401 | 0.406 | 0.403 | 0.411 | 0.404 | 0.406 | 0.620 | 0.580 |
| ETTm2 | **0.272** | **0.318** | 0.337 | 0.388 | 0.281 | 0.328 | 0.284 | 0.328 | 0.287 | 0.328 | 0.290 | 0.332 | 0.298 | 0.338 | 0.291 | 0.334 | 0.333 | 0.351 |
| electricity | **0.172** | **0.268** | 0.210 | 0.296 | 0.223 | 0.2327 | 0.206 | 0.294 | 0.215 | 0.293 | 0.183 | 0.282 | 0.179 | 0.278 | 0.175 | 0.270 | 0.313 | 0.384 |
| solar_AL | 0.244 | 0.271 | 0.327 | 0.397 | 0.244 | 0.349 | 0.268 | 0.322 | 0.356 | 0.350 | 0.257 | 0.292 | 0.268 | 0.298 | 0.239 | 0.280 | **0.197** | **0.244** |
| traffic | 0.469 | 0.292 | 0.626 | 0.386 | 0.500 | **0.287** | 0.556 | 0.365 | 0.624 | 0.375 | 0.510 | 0.348 | 0.506 | 0.335 | **0.462** | 0.307 | 0.640 | 0.348 |
| weather | **0.241** | **0.270** | 0.266 | 0.318 | 0.248 | 0.275 | 0.249 | 0.278 | 0.269 | 0.288 | 0.246 | 0.276 | 0.261 | 0.284 | 0.252 | 0.277 | 0.273 | 0.291 |
| 1st count | **5** | **4** | 0 | 0 | 0 | 1 | 0 | 1 | 0 | 1 | 0 | 0 | 1 | 1 | 1 | 0 | 1 | 1 |

## 4.3 ABLATION STUDY

To verify the effectiveness of each TIM component, we conducted a detailed ablation study on the proposed feature/time/resolution decomposition paradigm. The results of the ablation experiments are presented in Table 2. The prefix "wo" (now as a subscript) indicates "without," signifying the exclusion of specific model components during evaluation. The best results are highlighted in **bold red**, while the second-best performance is underlined in blue, providing a clear comparison of the relative effectiveness of different model configurations.

The ablation study results demonstrate that each component is essential. Notably, the Time and Res modules share the same architecture but differ in their operational sequence and input matrices in the ablation experiments, namely $Time_{wo}$ and $Res_{wo}$. Specifically, in $Time_{wo}$, the model learns temporal transitions across transposed multivariate time slices, whereas in $Res_{wo}$, it processes univariate time series as tokens to capture multivariate relationships.

Within the Feat module, each temporal token embeds multiple variables, encapsulating potential delayed events and distinct physical measurements. However, this approach may face challenges in capturing variate-specific representations, potentially leading to ineffective attention maps as the model prematurely learns complex latent spaces.

Table 2: Ablation Study

| TIM | | Ours | | $Time_{wo}$ | | $Res_{wo}$ | | $Feat_{wo}$ | |
|---|---|---|---|---|---|---|---|---|---|
| pred_len | | mse | mae | mse | mae | mse | mae | mse | mae |
| ETTh1 | 96 | **0.367** | **0.391** | 0.379 | 0.398 | 0.379 | 0.397 | 0.378 | 0.396 |
| | 192 | **0.424** | **0.425** | 0.438 | 0.428 | 0.436 | 0.427 | 0.433 | 0.426 |
| | 336 | **0.472** | **0.446** | 0.493 | 0.459 | 0.481 | 0.449 | 0.482 | 0.451 |
| | 720 | **0.471** | **0.469** | 0.497 | 0.477 | 0.494 | 0.478 | 0.492 | 0.476 |
| | AVG | **0.436** | **0.435** | 0.452 | 0.440 | 0.448 | 0.438 | 0.446 | 0.437 |
| ETTh2 | 96 | **0.289** | **0.342** | 0.291 | 0.343 | 0.292 | 0.344 | 0.292 | 0.344 |
| | 192 | **0.374** | **0.393** | 0.377 | 0.394 | 0.375 | 0.394 | 0.377 | 0.394 |
| | 336 | 0.419 | **0.430** | 0.418 | 0.431 | 0.417 | 0.430 | **0.417** | 0.430 |
| | 720 | **0.427** | **0.444** | 0.431 | 0.446 | 0.432 | 0.447 | 0.430 | 0.446 |
| | AVG | **0.377** | **0.402** | 0.379 | 0.404 | 0.379 | 0.404 | 0.379 | 0.403 |
| ETTm1 | 96 | **0.315** | **0.357** | 0.320 | 0.360 | 0.318 | 0.357 | 0.327 | 0.365 |
| | 192 | **0.361** | 0.383 | 0.366 | 0.385 | 0.361 | **0.381** | 0.364 | 0.384 |
| | 336 | **0.386** | **0.402** | 0.412 | 0.411 | 0.397 | 0.405 | 0.401 | 0.408 |
| | 720 | 0.469 | 0.446 | 0.495 | 0.452 | 0.456 | **0.441** | **0.454** | 0.442 |
| | AVG | **0.382** | 0.397 | 0.398 | 0.402 | 0.383 | **0.396** | 0.387 | 0.400 |
| ETTm2 | 96 | 0.172 | **0.253** | 0.176 | 0.259 | **0.170** | 0.254 | 0.175 | 0.258 |
| | 192 | **0.233** | **0.294** | 0.234 | 0.297 | 0.238 | 0.298 | 0.238 | 0.299 |
| | 336 | **0.292** | **0.333** | 0.295 | 0.337 | 0.299 | 0.338 | 0.301 | 0.339 |
| | 720 | **0.391** | **0.392** | 0.400 | 0.398 | 0.395 | 0.395 | 0.398 | 0.396 |
| | AVG | **0.272** | **0.318** | 0.276 | 0.323 | 0.276 | 0.321 | 0.278 | 0.323 |
| electricity | 96 | **0.144** | **0.241** | 0.156 | 0.255 | 0.152 | 0.253 | 0.169 | 0.269 |
| | 192 | **0.164** | **0.259** | 0.174 | 0.271 | 0.170 | 0.269 | 0.181 | 0.275 |
| | 336 | **0.173** | **0.271** | 0.190 | 0.289 | 0.186 | 0.287 | 0.197 | 0.290 |
| | 720 | **0.205** | **0.301** | 0.219 | 0.312 | 0.213 | 0.311 | 0.234 | 0.320 |
| | AVG | **0.172** | **0.268** | 0.185 | 0.282 | 0.180 | 0.280 | 0.195 | 0.288 |
| traffic | 96 | **0.447** | **0.277** | 0.473 | 0.313 | 0.474 | 0.306 | 0.492 | 0.311 |
| | 192 | **0.458** | **0.287** | 0.474 | 0.317 | 0.482 | 0.316 | 0.506 | 0.326 |
| | 336 | **0.471** | **0.292** | 0.482 | 0.317 | 0.492 | 0.320 | 0.519 | 0.332 |
| | 720 | **0.503** | **0.310** | 0.520 | 0.344 | 0.538 | 0.342 | 0.554 | 0.343 |
| | AVG | **0.469** | **0.292** | 0.487 | 0.323 | 0.497 | 0.321 | 0.518 | 0.330 |

Conversely, in the Time module, the time points of individual series are embedded into variate tokens, facilitating the capture of multivariate correlations. This design enables the $Res_{wo}$ configuration to achieve performance that is second only to the full TIM model, demonstrating its effective-

ness in enhancing multivariate analysis capabilities. Previous studies have suggested that tailoring model architectures specifically for datasets can lead to overfitting issues Li et al. (2024). However, our ablation experiments demonstrate that, across the majority of benchmarks, our TIM model, as a unified entity, exhibits optimal performance, thereby validating the efficacy of our novel decomposition framework. This underscores the indivisibility of its components, each contributing uniquely and synergistically to the overall performance.

## 4.4 MODEL EFFICIENCY

We undertake a comparative analysis of the operational memory consumption and execution time against the most recent state-of-the-art models during the training phase. Our findings consistently reveal that TIM exhibits remarkable efficiency advantages, both in terms of GPU memory utilization and running time, showcasing its favourable performance characteristics. Figure 3 shows that the horizontal axis of the chart employs Mean Squared Error (MSE) as its metric, while the vertical axis represents the logarithmically transformed number of model parameters. Despite having a comparable number of model parameters to other state-of-the-art approaches (SOTAs), the model significantly outperforms them in predictive performance. In this chart, each model is distinguished based on its prediction length ($pred\_len$), and the size of the points represents their Float Operations Per Second (FLOPs), which is a measure of computational performance. Furthermore, TIM stands out as a purely Multi-Layer Perceptron (MLP) architecture that successfully balances efficiency and performance. Unlike transformer-based models, which often require substantial computational resources and memory, TIM demonstrates remarkable proficiency in managing these demands with a more streamlined and efficient design.

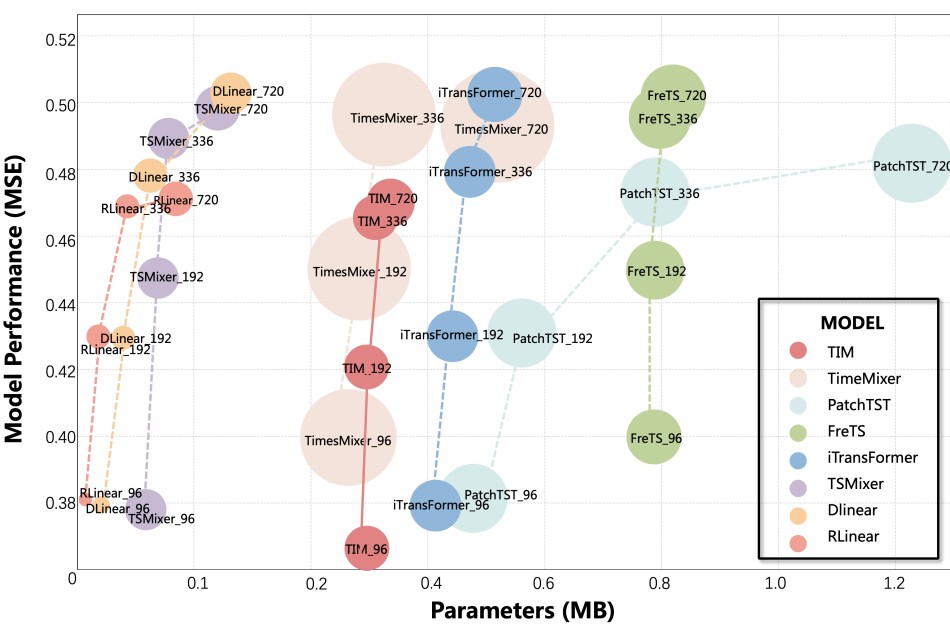

Figure 3: **Parameters** vs **Model performance (MSE)**. We reported the experiment This figure presents the experimental results for our models across various prediction lengths (pred_len) on the ETTh1 dataset. Notably, our all-MLP TIM has achieved SOTA performance while possessing a significantly smaller number of parameters compared to transformer-based models. The horizontal axis represents the logarithmic scale of model parameters (MB), and the vertical axis indicates the model performance measured by Mean Squared Error (MSE). For clarity in presentation, we applied a square root transformation to the model's parameter size, expressed in megabytes (MB).

## 5 CONCLUSION AND FUTURE WORK

In this paper, we introduced TIM, a model that achieves state-of-the-art performance in long-term time series forecasting while maintaining low computational complexity and resource efficiency. Our novel feature/time/resolution decomposition paradigm enables effective modelling of multi-variate interactions with minimal computational overhead, making the model particularly suitable for scenarios with limited resources.

While TIM demonstrates strong performance across various benchmarks, particularly due to its low-complexity design, further improvements can be made to capture more complex multivariate relationships. Future work will focus on refining the model's ability to handle these intricate inter-actions, without compromising its efficiency. By doing so, we aim to enhance both the predictive power and the practical applicability of the model in diverse real-world settings.

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

# A APPENDIX

## A.1 EXPERIMENT SETTING

To ensure a fair comparison across all models on a uniform platform (the time-series-library), we
have standardized the parameters. Specifically, we have fixed the input dimension at 96 and varied
the prediction horizon for time series forecasting, with lengths including 96, 192, 336, and 720. The
batch size was set to 32, the learning rate to 1e-3, the model dimension ($d\_model$) to 512, and the
dropout rate to 0.1.

## A.2 MAIN RESULT

Table 3: Multivariate forecasting results with prediction lengths in {96, 192, 336, 720} for eight benchmark datasets and fixed lookback length 96. Our proposed model TIM has achieved state-of-the-art (SOTA) performance on 25 tasks when evaluated using the Mean Squared Error (MSE) metric and on 21 tasks when assessed based on the Mean Absolute Error (MAE) metric. TIM exhibits robust performance across diverse benchmarks, which is particularly attributed to its low complexity and cross layer design. However, further enhancements can be implemented to capture better intricate multivariate relationships, especially in datasets with numerous variables and long time series.

| Models | | TIM Ours | | DLinear 2023 | | PatchTST 2023 | | FreTS 2024 | | RLinear 2023 | | TSMixer 2023 | | TimeMixer 2024 | | iTransFormer 2024 | | TimesNet 2023 | |
|---|---|---|---|---|---|---|---|---|---|---|---|---|---|---|---|---|---|---|---|
| Metric | | mse | mae | mse | mae | mse | mae | mse | mae | mse | mae | mse | mae | mse | mae | mse | mae | mse | mae |
| ETTh1 | 96 | 0.367 | 0.391 | 0.386 | 0.399 | 0.383 | 0.402 | 0.400 | 0.409 | 0.385 | 0.393 | 0.384 | 0.403 | 0.408 | 0.413 | 0.384 | 0.403 | 0.408 | 0.426 |
| | 192 | 0.424 | 0.425 | 0.434 | 0.428 | 0.435 | 0.431 | 0.455 | 0.440 | 0.436 | 0.422 | 0.444 | 0.435 | 0.457 | 0.442 | 0.434 | 0.431 | 0.496 | 0.475 |
| | 336 | 0.472 | 0.446 | 0.482 | 0.460 | 0.470 | 0.452 | 0.496 | 0.460 | 0.476 | 0.442 | 0.491 | 0.460 | 0.505 | 0.467 | 0.482 | 0.457 | 0.512 | 0.484 |
| | 720 | 0.471 | 0.469 | 0.504 | 0.502 | 0.479 | 0.476 | 0.506 | 0.481 | 0.478 | 0.467 | 0.505 | 0.485 | 0.492 | 0.478 | 0.491 | 0.482 | 0.708 | 0.580 |
| | AVG | 0.434 | 0.433 | 0.452 | 0.447 | 0.440 | 0.442 | 0.464 | 0.447 | 0.443 | 0.431 | 0.456 | 0.446 | 0.465 | 0.450 | 0.448 | 0.443 | 0.531 | 0.491 |
| ETTh2 | 96 | 0.289 | 0.342 | 0.329 | 0.384 | 0.292 | 0.344 | 0.298 | 0.348 | 0.290 | 0.340 | 0.304 | 0.353 | 0.293 | 0.342 | 0.302 | 0.352 | 0.343 | 0.378 |
| | 192 | 0.374 | 0.393 | 0.435 | 0.448 | 0.373 | 0.395 | 0.382 | 0.399 | 0.378 | 0.395 | 0.402 | 0.409 | 0.375 | 0.394 | 0.379 | 0.399 | 0.449 | 0.432 |
| | 336 | 0.419 | 0.430 | 0.563 | 0.526 | 0.417 | 0.431 | 0.426 | 0.436 | 0.430 | 0.439 | 0.444 | 0.445 | 0.398 | 0.424 | 0.418 | 0.430 | 0.468 | 0.459 |
| | 720 | 0.427 | 0.444 | 0.775 | 0.634 | 0.434 | 0.450 | 0.448 | 0.457 | 0.442 | 0.453 | 0.436 | 0.450 | 0.406 | 0.432 | 0.427 | 0.447 | 0.457 | 0.467 |
| | AVG | 0.377 | 0.402 | 0.526 | 0.498 | 0.379 | 0.405 | 0.448 | 0.457 | 0.385 | 0.407 | 0.396 | 0.414 | 0.368 | 0.398 | 0.382 | 0.407 | 0.429 | 0.434 |
| ETTm1 | 96 | 0.315 | 0.357 | 0.345 | 0.371 | 0.377 | 0.424 | 0.358 | 0.394 | 0.350 | 0.368 | 0.321 | 0.361 | 0.333 | 0.368 | 0.337 | 0.371 | 0.429 | 0.454 |
| | 192 | 0.361 | 0.383 | 0.383 | 0.394 | 0.417 | 0.439 | 0.399 | 0.411 | 0.388 | 0.386 | 0.370 | 0.388 | 0.376 | 0.393 | 0.376 | 0.388 | 0.593 | 0.572 |
| | 336 | 0.386 | 0.402 | 0.414 | 0.414 | 0.465 | 0.466 | 0.433 | 0.439 | 0.419 | 0.406 | 0.415 | 0.414 | 0.408 | 0.418 | 0.423 | 0.414 | 0.679 | 0.601 |
| | 720 | 0.469 | 0.446 | 0.474 | 0.453 | 0.517 | 0.501 | 0.538 | 0.509 | 0.480 | 0.440 | 0.497 | 0.461 | 0.493 | 0.464 | 0.480 | 0.449 | 0.780 | 0.692 |
| | AVG | 0.382 | 0.397 | 0.404 | 0.408 | 0.444 | 0.457 | 0.432 | 0.438 | 0.409 | 0.400 | 0.401 | 0.406 | 0.403 | 0.411 | 0.404 | 0.406 | 0.620 | 0.580 |
| ETTm2 | 96 | 0.172 | 0.253 | 0.186 | 0.282 | 0.176 | 0.261 | 0.180 | 0.262 | 0.182 | 0.265 | 0.183 | 0.267 | 0.183 | 0.267 | 0.184 | 0.268 | 0.188 | 0.268 |
| | 192 | 0.233 | 0.294 | 0.270 | 0.347 | 0.242 | 0.305 | 0.247 | 0.306 | 0.247 | 0.306 | 0.249 | 0.309 | 0.260 | 0.318 | 0.252 | 0.312 | 0.288 | 0.324 |
| | 336 | 0.292 | 0.333 | 0.362 | 0.414 | 0.303 | 0.344 | 0.304 | 0.342 | 0.309 | 0.344 | 0.310 | 0.347 | 0.309 | 0.346 | 0.317 | 0.352 | 0.343 | 0.360 |
| | 720 | 0.391 | 0.392 | 0.527 | 0.507 | 0.402 | 0.402 | 0.406 | 0.402 | 0.408 | 0.400 | 0.417 | 0.407 | 0.438 | 0.420 | 0.411 | 0.405 | 0.512 | 0.450 |
| | AVG | 0.272 | 0.318 | 0.337 | 0.388 | 0.281 | 0.328 | 0.284 | 0.328 | 0.287 | 0.328 | 0.290 | 0.332 | 0.298 | 0.338 | 0.291 | 0.334 | 0.333 | 0.351 |
| electricity | 96 | 0.144 | 0.241 | 0.195 | 0.277 | 0.206 | 0.309 | 0.184 | 0.271 | 0.198 | 0.274 | 0.156 | 0.258 | 0.153 | 0.253 | 0.146 | 0.244 | 0.351 | 0.405 |
| | 192 | 0.164 | 0.259 | 0.194 | 0.280 | 0.214 | 0.321 | 0.188 | 0.277 | 0.198 | 0.277 | 0.174 | 0.274 | 0.169 | 0.270 | 0.162 | 0.256 | 0.293 | 0.375 |
| | 336 | 0.173 | 0.271 | 0.208 | 0.297 | 0.218 | 0.325 | 0.203 | 0.294 | 0.212 | 0.293 | 0.187 | 0.288 | 0.184 | 0.285 | 0.180 | 0.274 | 0.290 | 0.373 |
| | 720 | 0.205 | 0.301 | 0.243 | 0.330 | 0.254 | 0.352 | 0.248 | 0.335 | 0.254 | 0.326 | 0.216 | 0.309 | 0.209 | 0.305 | 0.213 | 0.305 | 0.317 | 0.383 |
| | AVG | 0.172 | 0.268 | 0.210 | 0.296 | 0.223 | 0.2327 | 0.206 | 0.294 | 0.215 | 0.293 | 0.183 | 0.282 | 0.179 | 0.278 | 0.175 | 0.270 | 0.313 | 0.384 |
| solar_AL | 96 | 0.213 | 0.241 | 0.285 | 0.372 | 0.223 | 0.328 | 0.250 | 0.308 | 0.305 | 0.329 | 0.214 | 0.264 | 0.234 | 0.279 | 0.203 | 0.256 | 0.189 | 0.257 |
| | 192 | 0.234 | 0.266 | 0.316 | 0.393 | 0.246 | 0.353 | 0.268 | 0.328 | 0.344 | 0.348 | 0.257 | 0.292 | 0.277 | 0.306 | 0.233 | 0.271 | 0.193 | 0.234 |
| | 336 | 0.261 | 0.287 | 0.352 | 0.413 | 0.260 | 0.365 | 0.285 | 0.336 | 0.386 | 0.364 | 0.280 | 0.307 | 0.284 | 0.307 | 0.266 | 0.304 | 0.200 | 0.238 |
| | 720 | 0.267 | 0.289 | 0.355 | 0.411 | 0.246 | 0.350 | 0.269 | 0.315 | 0.389 | 0.358 | 0.278 | 0.304 | 0.278 | 0.300 | 0.254 | 0.286 | 0.207 | 0.248 |
| | AVG | 0.244 | 0.271 | 0.327 | 0.397 | 0.244 | 0.349 | 0.268 | 0.322 | 0.356 | 0.350 | 0.257 | 0.292 | 0.268 | 0.298 | 0.239 | 0.280 | 0.197 | 0.244 |
| traffic | 96 | 0.447 | 0.277 | 0.650 | 0.398 | 0.475 | 0.277 | 0.542 | 0.357 | 0.646 | 0.386 | 0.487 | 0.338 | 0.472 | 0.316 | 0.427 | 0.299 | 0.593 | 0.333 |
| | 192 | 0.458 | 0.287 | 0.599 | 0.371 | 0.489 | 0.278 | 0.537 | 0.358 | 0.599 | 0.362 | 0.496 | 0.338 | 0.494 | 0.328 | 0.451 | 0.302 | 0.631 | 0.349 |
| | 336 | 0.471 | 0.292 | 0.607 | 0.375 | 0.500 | 0.291 | 0.553 | 0.363 | 0.607 | 0.366 | 0.514 | 0.349 | 0.518 | 0.347 | 0.464 | 0.304 | 0.664 | 0.353 |
| | 720 | 0.503 | 0.310 | 0.648 | 0.398 | 0.535 | 0.302 | 0.590 | 0.380 | 0.645 | 0.385 | 0.541 | 0.368 | 0.540 | 0.350 | 0.506 | 0.324 | 0.673 | 0.359 |
| | AVG | 0.469 | 0.292 | 0.626 | 0.386 | 0.500 | 0.287 | 0.556 | 0.365 | 0.624 | 0.375 | 0.510 | 0.348 | 0.506 | 0.335 | 0.462 | 0.307 | 0.640 | 0.348 |
| weather | 96 | 0.155 | 0.200 | 0.196 | 0.255 | 0.165 | 0.211 | 0.167 | 0.213 | 0.193 | 0.232 | 0.159 | 0.208 | 0.160 | 0.207 | 0.168 | 0.211 | 0.194 | 0.233 |
| | 192 | 0.204 | 0.246 | 0.238 | 0.297 | 0.212 | 0.253 | 0.241 | 0.272 | 0.236 | 0.268 | 0.214 | 0.254 | 0.226 | 0.265 | 0.214 | 0.254 | 0.240 | 0.270 |
| | 336 | 0.262 | 0.289 | 0.283 | 0.333 | 0.268 | 0.292 | 0.269 | 0.295 | 0.288 | 0.304 | 0.273 | 0.294 | 0.286 | 0.307 | 0.273 | 0.296 | 0.292 | 0.307 |
| | 720 | 0.345 | 0.344 | 0.348 | 0.385 | 0.346 | 0.344 | 0.346 | 0.346 | 0.359 | 0.350 | 0.349 | 0.348 | 0.372 | 0.358 | 0.351 | 0.347 | 0.364 | 0.353 |
| | AVG | 0.241 | 0.270 | 0.266 | 0.318 | 0.248 | 0.275 | 0.249 | 0.278 | 0.269 | 0.288 | 0.246 | 0.276 | 0.261 | 0.284 | 0.252 | 0.277 | 0.273 | 0.291 |
| 1st count | | 25 | 21 | 0 | 0 | 2 | 5 | 0 | 0 | 0 | 6 | 0 | 0 | 3 | 3 | 5 | 1 | 5 | 4 |

## A.3 CODE OF ETHICS

We have read and understood the ICLR Code of Ethics, as outlined on the conference website. We fully acknowledge the importance of adhering to these ethical guidelines throughout all aspects of my participation in ICLR, including paper submission, reviewing, and discussions.

