# OpenReview forum: "TIM: Interpretable Modelling of Complex Temporal Interactions in Multivariate Networks"
_ICLR.cc/2025/Conference — ICLR 2025 Conference Withdrawn Submission_

### Official Review · Reviewer_6j4x · 2024-10-28

**Soundness:** 4
**Presentation:** 3
**Contribution:** 3
**Rating:** 3
**Confidence:** 3

**Summary:**

The paper introduces TIM, a novel multivariate time series forecasting method that leverages a cross-layer architecture to decompose time series predictions into temporal features, multivariate interaction features, and residual components. TIM employs a linear self-attention mechanism within an all-MLP model to enhance the learning of feature interactions and capture temporal transitions between domains.

**Strengths:**

1. TIM's feature/time/resolution decomposition paradigm is a significant innovation that enables effective modeling of multivariate interactions with minimal computational overhead.
2. TIM maintains linear computational complexity, a substantial advantage over Transformer-based models that suffer from quadratic complexity. The model incorporates mechanisms that enhance interpretability while providing robust performance across real-world time series data.
3. The paper presents a thorough experimental evaluation on multiple benchmarks, showcasing TIM's superiority in both accuracy and efficiency.

**Weaknesses:**

1. While TIM demonstrates strong performance, there is room for improvement in capturing more complex multivariate relationships, especially in datasets with numerous variables and long time series. In the ablation study, it is shown that different modules favor different datasets. For example, Feat plays a more important role in ETTm2, electricity, and traffic, which hasn't been fully and clearly explained in the paper.

2. The meaning of each module is well illustrated in the paper. But in Algorithm2 and Figure2, it's demonstrated that these three modules show a certain degree of sequential dependency. Is there any chance that a slight change in module order would drastically affect the results? This calls for a detailed illustration of the sequential design of these three modules.

**Questions:**

As is commented in Weakness.

---

### Official Review · Reviewer_RNKQ · 2024-11-01

**Soundness:** 2
**Presentation:** 2
**Contribution:** 2
**Rating:** 3
**Confidence:** 4

**Summary:**

The paper introduces TIM, a novel multivariate time series forecasting model. TIM integrates cross-layer architecture and linear self-attention to decompose and model temporal features, multivariate interactions, and residual components. By addressing limitations of Transformer and MLP models, such as high computational cost and difficulty in capturing complex dependencies. TIM presents an MLP-based approach that enhances forecasting accuracy while maintaining computational efficiency. Experimental results across benchmark datasets indicate that TIM surpasses SOTA models in both accuracy and efficiency.

**Strengths:**

Originality: TIM introduces a MLP-based approach incorporating cross-layer and linear attention mechanisms.

Quality: The model is evaluated across multiple datasets and prediction horizons, with ablation studies verifying the impact of each component.

Clarity: The structure is clear, with a logical flow through the introduction, methodology, experiments, and conclusions.

Significance: TIM addresses a problem in multivariate time series forecasting by achieving a balance between accuracy and computational efficiency.

**Weaknesses:**

While the article focuses on providing a detailed introduction to the framework, it lacks certain discussions on theory and significance. Firstly, the article's title emphasizes interpretability, but the entire paper discusses interpretability very little.
I understand that the components of the framework in the paper seem to be designed for interdependency or interactions, the authors need to clearly state what they have interpreted. A more explicit discussion is needed to discuss interaction -> interpretability.

Among the many models in time series, one often either optimizes performance or interpretability. The authors' approach of titling with interpretability but emphasizing prediction error throughout the paper confuses me. The paper does not compare with other time series interpretability articles within the related work section.

There are some other minor concepts that I think should be discussed in the article. They are listed in the 'Questions' part.

**Questions:**

1. Interpretability: The title emphasizes "interpretable" modeling, but the main paper lacks a discussion of interpretability. Could you clarify how interpretability is integrated within the TIM model?  Does the model offer insights into specific feature interactions or temporal dependencies?

2. There is considerable discussion about the attention mechanism, including Equation (1), which might cover too much conventional knowledge. Could some of this space be repurposed to focus on the novel contributions and unique aspects of the proposed model?

3. The paper suggests advantages for long-term time series but does not provide a specific validation for this claim. Would performance improve with increasing sequence length, and could this be demonstrated to highlight the model's strength in handling longer sequences?

4. The linear attention mechanism’s computational advantages are discussed, yet its theoretical justification, especially in terms of time complexity, could be elaborated upon.

---

### Official Review · Reviewer_J98b · 2024-11-03

**Soundness:** 2
**Presentation:** 2
**Contribution:** 2
**Rating:** 3
**Confidence:** 4

**Summary:**

The paper introduces TIM, an all-MLP model for multivariate time series forecasting designed to overcome the computational and performance limitations of Transformers and MLPs. The authors claim that TIM employs a cross-layer architecture to decompose predictions into temporal, multivariate, and residual components, using a linear self-attention mechanism to capture complex feature interactions and transitions efficiently. The experiments show that the model achieves state-of-the-art performance with linear computational complexity, demonstrating both accuracy and efficiency across experiments.

**Strengths:**

1. The experiments show that the proposed model is outperforming several baselines.
2. The empirical model efficiency seems better than others.

**Weaknesses:**

1. The authors did not provide an analysis on computational complexity, although an empirical efficiency comparison is provided. In my view the computational complexity analysis is needed as it's helpful for people to understand why the proposed model should be more efficient. Also, by having the complexity analysis, we can better understand in what scenarios, the model should perform faster than others. To me, I did not understand where the better efficiency comes from.
2. The experiments seem not very sufficient. The paper only includes 7 baselines. I feel a few more baselines should be compared. For example, another category of models TCN-based models such as ModernTCN [1].

Reference:

[1] ModernTCN: A Modern Pure Convolution Structure for General Time Series Analysis. ICLR 2024

**Questions:**

1. I am a bit confused that why the authors claimed that the model is all-MLP but still has a linear self-attention mechanism? If Line 7 of Algo 1 is referred to as the linear self-attention, it should probably be explicitly explained more.
2. I don't fully understand Algorithm 1, especially the part of residual and its relationship with y. Based on Line 3 and 4, $res = \hat W_i(X_i - W_iX_i - b_1) + b_2$. Then, because $W_i and \hat W_i$,  $b$ are learnable, residual can be rewritten as a new linear projection without dependence on $y$. If so, what is the propose of using $x_{cat} = cat(y,res)$? What would happen if we do not use the residual within the fusion architecture?

---

### Note · Authors · 2024-11-19

I have read and agree with the venue's withdrawal policy on behalf of myself and my co-authors.